# Mammalian SIRT6 Represses Invasive Cancer Cell Phenotypes through ATP Citrate Lyase (ACLY)-Dependent Histone Acetylation

**DOI:** 10.3390/genes12091460

**Published:** 2021-09-21

**Authors:** Wei Zheng, Luisa Tasselli, Tie-mei Li, Katrin F. Chua

**Affiliations:** 1Department of Medicine, Division of Endocrinology, Gerontology, and Metabolism, Stanford University School of Medicine, Stanford, CA 94305, USA; luisatasselli@gmail.com (L.T.); tiemei.li2@gmail.com (T.-m.L.); 2Geriatric Research, Education, and Clinical Center, VA Palo Alto Health Care System, Palo Alto, CA 94304, USA

**Keywords:** SIRT6, Sirtuin, ACLY, histone acetylation, acetyl-CoA, chromatin, cancer, gene expression

## Abstract

The modulation of dynamic histone acetylation states is key for organizing chromatin structure and modulating gene expression and is regulated by histone acetyltransferase (HAT) and histone deacetylase (HDAC) enzymes. The mammalian SIRT6 protein, a member of the Class III HDAC Sirtuin family of NAD+-dependent enzymes, plays pivotal roles in aging, metabolism, and cancer biology. Through its site-specific histone deacetylation activity, SIRT6 promotes chromatin silencing and transcriptional regulation of aging-associated, metabolic, and tumor suppressive gene expression programs. ATP citrate lyase (ACLY) is a nucleo-cytoplasmic enzyme that produces acetyl coenzyme A (acetyl-CoA), which is the required acetyl donor for lysine acetylation by HATs. In addition to playing a central role in generating cytosolic acetyl-CoA for de novo lipogenesis, a growing body of work indicates that ACLY also functions in the nucleus where it contributes to the nutrient-sensitive regulation of nuclear acetyl-CoA availability for histone acetylation in cancer cells. In this study, we have identified a novel function of SIRT6 in controlling nuclear levels of ACLY and ACLY-dependent tumor suppressive gene regulation. The inactivation of SIRT6 in cancer cells leads to the accumulation of nuclear ACLY protein and increases nuclear acetyl-CoA pools, which in turn drive locus-specific histone acetylation and the expression of cancer cell adhesion and migration genes that promote tumor invasiveness. Our findings uncover a novel mechanism of SIRT6 in suppressing invasive cancer cell phenotypes and identify acetyl-CoA responsive cell migration and adhesion genes as downstream targets of SIRT6.

## 1. Introduction

Acetyl-coenzyme A (acetyl-CoA) is a key metabolic intermediate that links the metabolism, signaling, and epigenetics. It is the acetyl donor for protein acetylation reactions, including histone lysine acetylation, which play important roles in chromatin dynamics and gene regulation [1,2,3]. The equilibrium of histone acetylation is maintained largely by the opposing activities of histone acetyltransferase (HATs) and histone deacetylase (HDACs) enzymes [4], and histone acetylation levels at certain histone residues are frequently deregulated in tumors [5,6,7]. Intriguingly, accumulating evidence indicates that histone acetylation is sensitive to the availability of acetyl-CoA, the sole donor of acetyl groups and an obligate cofactor for HATs, and suggests that the nucleo-cytoplasmic distribution and the abundance of nuclear pools of acetyl-CoA can directly impact the histone acetylation by HATs [1,8,9,10,11]. Two nucleo-cytoplasmic enzymes have been reported to play key roles in maintaining nuclear acetyl-CoA abundance: acetyl-CoA synthetase 2 (ACSS2), which converts acetate to acetyl-CoA and directly regulates histone acetylation in differentiating neurons [12,13]; and ATP citrate lyase (ACLY), which catalyzes the conversion of citrate to oxaloacetate and acetyl-CoA [11,14]. The deregulation of ACLY is linked to various diseases including cancers, due to its central role in nutrient metabolism and biosynthesis [15]. However, ACLY-mediated nuclear acetyl-CoA production was only recently shown to be an important mechanism in controlling HAT-dependent histone acetylation, which impacts specific gene expression profiles during tumorigenesis [16,17,18,19], highlighting the role of ACLY as an important epigenetic regulator.

SIRT6 is a Class III HDAC in the Sirtuin family of NAD+-dependent enzymes that have fundamental roles in aging, metabolism, and stress responses [20,21]. SIRT6 is a complex enzyme with multiple reported enzymatic mechanisms including catalyzing protein de-acetylation, de-fatty-acylation, and promoting mono-ADP-ribosylation [22,23,24]. SIRT6 has been most studied for its site-specific de-acetylase activity on key histone H3 lysine substrates, including H3K9ac, H3K18ac, H3K56ac, and H3K27ac [25,26,27,28,29], which contributes to the dynamic regulation of chromatin and gene expression. Through histone deacetylation at diverse target genes, SIRT6 is a central regulator of transcriptional programs in aging, cellular metabolism, and cancer biology [24,30,31]. Importantly, SIRT6 is implicated in tumor suppression by repressing transcription programs linked to cell proliferation, inflammation, ribosomal biogenesis, metabolic reprogramming, and genomic instability [27,32,33,34,35,36,37], and accordingly, down-regulated SIRT6 expression is observed in multiple human cancers [34,38]. Notably, SIRT6 is also implicated in cancer cell migration and metastasis; however, controversy remains over whether it promotes or prevents these processes [39,40,41], with conflicting reports from studies involving different cancer cell types.

In a SIRT6 interactome study, we previously identified ACLY as a SIRT6-interacting protein [42]. Both SIRT6 and ACLY are implicated in the cellular responses to nutrient availability and impact metabolic and neoplastic transcriptional programs via the regulation of site-specific histone acetylation states, prompting us to investigate a potential functional interplay between these proteins. Previously, ACLY-dependent acetyl-CoA production was shown to promote cell migration and extracellular matrix (ECM) adhesion phenotypes that contribute to the invasiveness of glioblastoma multiforme (GBM) tumor cells. ACLY achieves these effects by impacting levels of H3K27ac, and potentially other histone sites, at the specific loci driving expression of a small set of genes that control cell adhesion, migration, and integrin signaling [2,17]. Among the genes that are most sensitive to acetyl-CoA abundance, Platelet Derived Growth Factor Receptor A (PDGFRA), a tyrosine kinase that is frequently dysregulated and promotes cell survival and metastasis in cancers [43,44,45], showed dramatic acetyl-CoA dependent changes in H3K27 acetylation around its transcriptional start site (TSS) [17]. Since H3K27ac is a SIRT6 substrate [26], these findings raised the intriguing possibility that there may be a mechanistic link between SIRT6 and ACLY, to maintain nuclear acetyl-CoA pools in a coordinated manner and regulate the expression of genes that contribute to the cell migration and adhesion phenotypes.

In this study, we show that SIRT6 deficiency increases the nuclear ACLY protein, nuclear acetyl-CoA abundance, and locus-specific histone acetylation, which in turn drive an up-regulation of PDGFRA and other genes that contribute to invasive cancer cell adhesion and migration phenotypes. Inhibition of ACLY in SIRT6-deficient cells reversed the dysregulated expression and histone hyperacetylation and prevented the migration and adhesion phenotypes of SIRT6-deficient cancer cells. Together, our findings uncover a role of SIRT6 in suppressing malignant cancer cell properties and reveal a novel mechanism through which SIRT6 may interface with histone acetylation machinery to reinforce and fine tune its effects on histone acetylation dynamics in cancer cell gene regulation.

## 2. Materials and Methods

### 2.1. Cell Culture, Viral Infection, and Drug Treatment

Human U2OS cells and 293T cells (ATCC) were cultured in EMEM supplemented with 10% FBS and Pen/Strep. Human HCT116 cells (ATCC) were cultured in DMEM supplemented with 10% FBS, 2 mM L-Glutamine, and Pen/Strep. U2OS and HCT116 cells with SIRT6 knock-out or knockdown were generated by the lentiviral infection of CRISPR/Cas9 gRNA or shRNA targeting SIRT6, respectively, as previously described [27]. U2OS cells stably expressing Flag-tagged or HA-tagged human ACLY, SIRT6, and SIRT6 catalytic mutants were generated by retroviral transduction as described previously [27]. Control or ACLY (L-004915-00-0005, Dharmacon, Lafayette, CO, USA) siRNAs were transfected at 50 nM using Dharmafect reagent following the manufacturer’s instructions (Dharmacon), and samples were analyzed at 72 h post-transfection. ACLY inhibitor BMS-303141 (Sigma-Aldrich) was added to cells at 50 µM for 24 h followed by subsequent assays and analyses.

### 2.2. Co-Immunoprecipitation (IP)

Cell lysates were prepared with buffer A (50 mM Tris-HCL pH: 7.4; 250 mM NaCL; 0.5% TritonX-100; 10% Glycerol) as described previously [46]. A total of 5 mg of total protein was used for immunoprecipitation with SIRT6, ACLY, or rabbit anti-mouse IgG antibody overnight. The immunoprecipitated complex was incubated with protein A/G beads (Thermo Fisher, Waltham, MA, USA) for 3 h followed by washing with buffer A 3 times. The samples were eluted from beads in the Laemmli buffer at 98 °C and used for a Western blot analysis. Flag-immunoprecipitation was carried out with 2 mg of total protein, and immunoprecipitated by anti-FLAG M2 affinity gel (A2220, Sigma-Aldrich, St. Louis, MO, USA).

### 2.3. Acetyl-CoA Measurement

To extract and quantify nuclear acetyl-CoA, 5 × 10^6^ U2OS cells were harvested into cell lysis buffer (10 mM Tris pH 8, 1 mM KCl, 1.5 mM MgCl2, 1 mM DTT) and incubated on ice for 30 min. The nuclei were pelleted at 3000 g for 5 min and re-suspended immediately in Acetyl-CoA assay buffer provided in the PicoProbe acetyl-CoA Assay Kit (ab87546, Abcam). Samples were prepared and deproteinized according to the manufacturer’s instructions. The PicoProbe acetyl-CoA assay was performed in 96-well clear-bottom plates, and the resulting fluorescence was measured using the Synergy H1 Microplate Reader (BioTek Instruments, Winooski, VT, USA). Results were calculated and quantified as acetyl-CoA level normalized to the total protein.

### 2.4. RNA Extraction and QPCR

Total RNA was extracted from cells using the TRIzol reagent (Thermo Fisher, Waltham, MA, USA). cDNA was synthesized using the SuperScript IV First-Strand Synthesis System (Thermo Fisher, Waltham, MA, USA) according to manufacturer’s instructions. A quantitative real-time PCR analysis was performed on a Roche LightCycler 480 using SYBR green PCR Master Mix (Thermo Fisher, Waltham, MA, USA). Relative mRNA levels were normalized to GAPDH expression. Sequences of primers for the acetyl-CoA responsive genes are previously described [16]. Sequences of primers for the SIRT6 target glycolytic genes are listed in Appendix A.

### 2.5. Chromatin Immunoprecipitation (ChIP)

ChIPs in U2OS cells were performed as described previously [27]. Briefly, 5 × 10^6^ cells were crosslinked with 1% formaldehyde for 7 min at room temperature, quenched with 1.25 M glycine, washed in ice-cold PBS, and pellets were resuspended in the lysis buffer (50 mM Tris pH 8.0, 10 mM EDTA, 1% SDS). Cell lysates were sonicated and diluted in the RIPA buffer (10 mM Tris pH 7.5, 140 mM NaCl, 1 mM EDTA, 0.5 mM EGTA, 1% Triton, 0.1% SDS, 0.1% sodium deoxycholate). Antibodies against specific histone marks were coupled to Dynabeads protein A (Thermo Fisher, Waltham, MA, USA) for >2 h at 4 °C. Rabbit IgG antibody was used as the negative control in all the experiments. The antibody-bead complexes were added to sonicated lysates and rotated overnight at 4 °C. Beads were washed three times in RIPA, and DNA was eluted and reverse cross-linked at 65 °C for 2 h using the elution buffer (20 mM Tris pH 7.5, 5 mM EDTA, 50 mM NaCl, 1% SDS, 5 ug/mL proteinase K). Eluted DNA was recovered using a PCR purification kit (Qiagen, Hilden, Germany) and assayed by RT-qPCR on a LightCycler 480 using SYBR green Master Mix (Thermo Fisher, Waltham, MA, USA). The sequences of primers for the PDGFRA gene promoter were described previously [17]. Fold enrichment was calculated as % input and normalized to total H3.

### 2.6. Wound Healing Scratch Assay

U2OS cells were grown in 24-well tissue culture plates to 80–90% confluency in monolayers. A scratch was then made using 20 uL pipette tip on cell monolayers. Cells were washed twice with fresh media and replaced with media containing DMSO or 50 µM ACLY inhibitor. Cells were imaged immediately after creating the wound with a time lapse microscope using a 10× objective, and images were captured every 1 h for 24 h. Wound closure was then quantified using Image J software by calculating the distance between the wound edges from 8 images per treatment condition per time point.

### 2.7. Soft Agar Assay

The soft agar assay was performed in six-well plates. For each well, 2 mL complete medium containing 1% agarose was poured into the well and allowed to cool to room temperature to form a bottom layer. Then, 2 × 10^4^ HCT116 cells were mixed in 2 mL complete medium containing a final concentration of 0.4% agarose and poured on top of the bottom layer and incubated at 4 °C for 5–10 min before moving to 37 °C incubator. Next, 1 mL of complete medium was added to each well and the medium was replaced every 3 days. After 14 days, the colonies were counted and analyzed using a 4× objective lens.

### 2.8. Fibronectin Adhesion Assay

Before seeding the cells, 96-well plates were coated with 1% human plasma fibronectin (Millipore) at 4 °C overnight and blocked with fresh media at 37 °C for 30 min. Cells were treated with DMSO or the ACLY inhibitor for 24 h before seeding to the wells. For U2OS and HCT116 cells, 6 × 10^4^ cells were seeded per well and allowed to adhere in the CO_2_ incubator at 37 °C for 30 min. Wells were then washed three times with PBS and stained with 0.1% crystal violet solution for 30 min at room temperature. Crystal violet was washed with water, and the plate was air-dried overnight. Crystal violet was solubilized in 10% acetic acid solution, and the absorbance was read at 590 nm. Relative absorption was calculated by absorbance corrected to empty wells and the fold difference relative to the control cells treated with DMSO.

### 2.9. Antibodies

The following antibodies were used for ChIP: Histone H3 antibody (ab1791), Histone H3 (acetyl K9) antibody (ab10812), H3 tri methyl K9 antibody (ab8898), Histone H3 (acetyl K18) antibody (ab1191), Histone H3 (acetyl K27) antibody (ab4729), and Rabbit anti-IgG control (ab46540) from Abcam. Primary antibodies used for IP and Western blotting were: Anti-ATP citrate lyase antibody (ab157098), Anti-Fibrillarin antibody [38F3] (ab4566), Histone H3 antibody (ab1791), and Rabbit anti-IgG (ab46540) from Abcam; Anti-β-Actin Antibody (A5441) and Anti-Flag M2 monoclonal antibody (F1804) from Sigma-Aldrich; SIRT6 (D8D12) Rabbit mAb (CST12486) and PDGF Receptor α antibody (CST3164) from Cell Signaling; Acetyl-Histone H3 Antibody (06-599) and Anti-β-Tubulin Antibody clone AA2 (05-661) from Millipore.

## 3. Results

### 3.1. SIRT6 Deficiency Leads to Increased Levels of ACLY

In exploring a potential functional interaction of SIRT6 with ACLY, we were surprised to observe that ACLY protein levels were consistently elevated in SIRT6-deficient cancer cells (Figure 1A). This occurred following knock-down with multiple independent shRNAs or Crispr/Cas9 mediated knockout, excluding off-target effects. Moreover, ACLY protein levels were also increased in multiple tissues in SIRT6 knockout mice, including the brain and liver, demonstrating a physiologic role of SIRT6 in determining basal levels of ACLY expression in vivo (Figure 1B).

Because nuclear acetyl-CoA production by ACLY impacts histone acetylation and gene expression in malignant phenotypes [17,18], we performed sub-cellular fractionation assays to examine the effects of SIRT6 on ACLY levels in different sub-cellular compartments. This analysis revealed that SIRT6 deficiency leads to an increase in ACLY protein levels in both nuclear and chromatin-bound biochemical fractions (Figure 1C). The overexpression of wild-type SIRT6, but not a catalytically inactive mutant SIRT6 protein (SIRT6-H133Y), reversed the increased ACLY protein in SIRT6-deficient cells (Figure 1D), consistent with an underlying catalytic mechanism. By contrast, we found that SIRT6 interaction with the ACLY protein was independent of its enzymatic activity and histone association (Figure 1E), suggesting that the effect of SIRT6 on ACLY protein abundance occurs by a distinct mechanism. Notably, the levels of ACLY mRNA (Appendix A) and of a stably expressed recombinant Flag-tagged ACLY protein were both increased in SIRT6-deficient cells (Figure 1F), indicating that SIRT6 impacts ACLY abundance at both the transcriptional and post-transcriptional levels.

Next, we asked if the nuclear levels of acetyl-CoA are altered by the increased ACLY levels in SIRT6-deficient cells. An analysis of isolated nuclei from SIRT6 KO cells revealed a 26% increase (*p* = 0.03) in nuclear acetyl-CoA compared to the control cells (Figure 1G). Notably, this is comparable to the magnitude of nuclear acetyl-CoA changes that were previously shown to have functional effects on histone acetylation and gene expression [12]. Together, these data demonstrate that SIRT6 loss leads to increased levels of nuclear and chromatin-associated ACLY and nuclear acetyl-CoA, and suggest a novel mechanism whereby SIRT6 could reinforce its direct histone deacetylation activity by coordinately tamping down on histone acetylation.

### 3.2. ACLY-Dependent Up-Regulation of Acetyl-coA Responsive Genes in SIRT6-Deficient Cells

We next tested directly whether the increased abundance of nuclear ACLY and acetyl-CoA in SIRT6 KO cells leads to functional changes in histone acetylation and gene expression. We first examined mRNA expression of a group of genes that were previously shown to be highly sensitive to acetyl-Co-A abundance and that are involved in cancer cell migration and ECM adhesion [16]. Strikingly, we observed significant up-regulation of these genes in SIRT6 KO cells (Figure 2A and Appendix A). Among these genes, PDGFRA was most dramatically affected by SIRT6 loss (12-fold) and was also significantly up-regulated at the level of protein expression (Figure 2B and Appendix A). We have also observed a significant increase in the expression of known SIRT6 target glycolytic genes in the same SIRT6-deficient cells (Figure 2C and Appendix A) [34,35]. These data identify the acetyl-CoA responsive genes as novel downstream targets of SIRT6 and show that they are up-regulated by SIRT6-loss at levels comparable to or substantially greater than the previously described SIRT6 target genes. We next asked if the observed gene expression changes are due to the increased ACLY levels in SIRT6-deficient cells. Treatment with an ACLY inhibitor substantially reduced the up-regulated expression of PDGFRA in SIRT6 KO cells (Figure 2D and Appendix A) but had varying effects on the expression of the SIRT6 target glycolytic genes (Appendix A). For some of the glycolytic genes, such as LDHA, there was a trend towards reversal of the overexpression in SIRT6 KO cells that was statistically significant in some experiments, suggesting some contribution of ACLY to their regulation by SIRT6. Surprisingly, ACLY inhibition led to an unexpected increase in the expression of several other SIRT6 target glycolytic genes in both control and SIRT6 deficient cells, suggesting an additional link of ACLY to metabolic gene regulation that may be independent of SIRT6 (Appendix A). Together, our observations suggest that SIRT6-dependent changes in ACLY abundance preferentially affect genes involved in cancer cell migration and adhesion, and may also impact a subset of previously described SIRT6 target glycolytic genes.

To study the effects of SIRT6 and ACLY on locus-specific histone acetylation, we carried out Chromatin IP (ChIP) analyses at the PDGFRA and LDHA genes. This analysis revealed significant hyperacetylation of H3K9ac, H3K18ac, and H3K27ac in SIRT6-deficient cells at both the PDGFRA and LDHA upstream regulatory regions (Figure 2E,F). Notably, the hyperacetylation at PDGFRA was significantly reduced by ACLY inhibition, consistent with previous reports that PDGFRA expression and H3K27ac occupancy at the PDGFRA TSS are especially sensitive to acetyl-CoA availability and ACLY activity [17]. By contrast, ACLY inhibition did not significantly affect histone hyperacetylation on the LDHA promoter in SIRT6-deficient cells (Figure 2F). Together these observations suggest that SIRT6 may coordinate histone deacetylation at its direct target genes with an inverse regulation of ACLY-dependent histone acetylation at acetyl-CoA responsive genes, providing a multi-pronged repression of distinct transcriptional programs that contribute to cancer cell malignant phenotypes. 

### 3.3. ACLY-Dependent SIRT6 Regulation of Cancer Migration and Adhesion Phenotypes

Previous work showed that ACLY-dependent acetyl-CoA production promotes cancer cell adhesion and migration, properties that contribute to tumor invasiveness [17]. We therefore examined the effects of SIRT6 on the invasive properties of cancer cells using several independent assays and cancer cell types. In a soft agar colony formation assay of U2OS osteosarcoma cells, the control cells largely formed compact cell masses with relatively smooth edges (Figure 3A,B). By contrast, a vast majority (~90%) of SIRT6 KO U2OS cells had looser cell–cell contacts and formed cell masses with rougher, less clearly defined edges (Figure 3A,B), with cells at the colony periphery prone to escape from the local mass. This heterogeneous morphology and more aggressive migratory behavior are highly reminiscent of an invasive, metastatic phenotype, as previously described [47]. In an independent soft agar assay using HCT116 colon cancer cells, SIRT6 KO cells exhibited significantly increased anchorage independent colony formation (32% increase, *p* < 0.01) (Figure 3C), a hallmark of cancer cell malignancy. We also used the wound healing scratch assay to assess the migratory behavior of SIRT6 KO versus control U2OS cells. Confluent cell cultures were subjected to a scratch wound, and wound closure followed over 24 h. SIRT6 KO cells showed significantly faster wound closure than the control cells, and reconstitution with wild-type SIRT6 (WT), but not the catalytically inactive mutant SIRT6-HY protein, reversed the accelerated migration of the SIRT6 KO cells (Figure 3D,E). Importantly, treatment with the ACLY inhibitor significantly attenuated the accelerated wound healing of SIRT6 KO cells to levels comparable to ACLY inhibition in control cells (Figure 3F,G), providing evidence that the increased migration of SIRT6 KO cells is dependent on ACLY. Finally, the fibronectin adhesion assay was used to assay effects of SIRT6 and ACLY on cell adhesion to the extracellular matrix (ECM), which contributes to cancer cell invasiveness. SIRT6 deficient U2OS and HCT116 cells both showed significantly increased adhesion to fibronectin compared to control cells, which was reversed by ACLY inhibition (Figure 3H). Together, our observations demonstrate a novel tumor suppressive mechanism of SIRT6 on cancer cell adhesion and migration through the control of ACLY-dependent gene regulation.

## 4. Discussion

Here, we have characterized a functional interaction between SIRT6 and ACLY which plays essential roles in modulating nuclear acetyl-CoA abundance, locus-specific histone acetylation, and the expression of cancer cell adhesion and migration genes. SIRT6 has previously been shown to function in tumor suppression by reprogramming multiple gene expression pathways through histone deacetylation. Our findings here indicate that SIRT6 impacts histone acetylation levels not only by direct deacetylation, but also by tamping down on the nuclear abundance of ACLY and its production of acetyl-CoA. These effects of SIRT6 mirror those previously shown in a genome-wide study which identified a relatively small set of genes whose expression and TSS acetylation are highly sensitive to acetyl-CoA availability, and which promote cell migration and adhesion phenotypes [17]. Notably, the prior study of acetyl-CoA responsive genes only examined acetylation on H3K27ac. By contrast, our findings reveal effects of the SIRT6-ACLY axis on H3K9ac, H3K18ac, and H3K27ac, demonstrating that the modulation of acetylation levels by acetyl-CoA pools is not unique to the H3K27ac mark. A systematic analysis of the spectrum of histone acetylation marks that respond to acetyl-CoA availability will be an important goal for future study.

Our observations also suggest a model of SIRT6 function similar to the “feed-forward” loops of transcriptional regulation [48,49], in which an input (here SIRT6 activity) feeds into two processes (histone deacetylation and suppression of acetyl-CoA precursor for histone acetylation) that both contribute to maintaining the repression of cancer cell gene expression programs (Figure 4). One potential consequence that is proposed to arise from such feed-forward regulation is an inherent time delay that can buffer against noisy transcriptional inputs [48]. In the case of SIRT6, the two-pronged regulation of both histone deacetylation and ACLY-dependent histone acetylation may help maintain the stability of tumor suppressive gene expression programs and protect against potential effects of metabolic fluctuations on SIRT6 or ACLY activity. Our findings also uncover a new mechanism of SIRT6 in suppressing metastatic cancer cell phenotypes and identify acetyl-CoA responsive cell migration and adhesion genes as downstream targets of SIRT6.

In addition to its roles in cancer cell adhesion and migration, which we have focused on in this study, ACLY is the major enzyme catalyzing glucose dependent production of acetyl-CoA, and as such has pivotal roles in cancer cell metabolism by supporting protein acetylation, de novo lipid biosynthesis, and promoting glycolysis and the Warburg effect. These functions overlap substantially with the central functions of SIRT6 in controlling cancer cell metabolism; however, we found that the increased expression and histone hyperacetylation of SIRT6 target glycolytic genes that occurs in SIRT6-deficient cells was not significantly rescued by ACLY inhibition. This observation is consistent with the previous report that the potent regulation of H3K27ac at promoter TSSs is observed at a relatively small number of acetyl-CoA-up-regulated genes [17]. This specificity may reflect the selective, locus-specific activity of HATs that are impacted by the altered acetyl-CoA pools [50] and/or highly localized regulation of acetyl-CoA levels by ACLY [1,51].

Consistent with our findings, the up-regulation of ACLY levels have been implicated in promoting cancer malignancy and poor prognosis [52]. Conversely, the down-regulation of ACLY by miR-22 was shown to attenuate cancer cell proliferation and invasion, as well as promote cell apoptosis in different types of tumor cells including osteosarcoma, prostate, cervical, and lung cancers [53]. Besides regulating cancer metabolism, the importance of ACLY in mediating histone acetylation and gene expression profiles in cancer cells has also been recognized [11]. Interestingly, ACLY mRNA expression was reported to increase in Huh7 hepatocellular carcinoma cells with SIRT6 overexpression and decrease with miR122 which is an antagonist of SIRT6 [54]; by contrast, our findings here indicate that SIRT6 deficiency increases ACLY expression in U2OS cells at the level of mRNA and protein (Figure 1 and Appendix A). One intriguing possibility is that in some conditions or cell types as used in the previous studies [54], the SIRT6-dependent changes in ACLY protein levels may trigger compensatory negative feedback regulation on ACLY mRNA. A goal for future studies will be to examine the underlying mechanisms and physiologic conditions that contribute to the differing effects of SIRT6 on ACLY mRNA.

Our study of SIRT6 and ACLY was prompted by our prior identification of ACLY as a candidate SIRT6-interacting protein in a proteomic screen ([42], Figure 1E). However, the novel functions of SIRT6 in ACLY regulation described here appear to be independent of a physical ACLY–SIRT6 interaction, because they are dependent on SIRT6 catalytic activity whereas the ACLY–SIRT6 interaction is not (Figure 1D,E). An intriguing observation from our ACLY–SIRT6 interaction experiments is that the ACLY levels are similar in the IP of the SIRT6 S56Y mutant as in the WT and other mutant IPs, even though there is much less S56Y protein in the input (due to the lower expression of this mutant protein). This could indicate that the S56Y mutant interacts more strongly with ACLY, or alternatively, that the levels of SIRT6 protein (WT or mutant) are not limiting for the interaction with ACLY.

The specific mechanism(s) through which SIRT6 modulates ACLY protein abundance will also be an important question for future study. It has been reported that ACLY acetylation at lysine residues 540, 546, and 554 increases its stability, and that SIRT2, a SIRT6 family member, can deacetylate and destabilize ACLY in the cytoplasm [55]. It may be that analysis of nuclear ACLY protein acetylation will reveal if SIRT6-dependent deacetylation of ACLY contributes to the effects of SIRT6 on ACLY stability and abundance in cancer cells.

Our findings also provide the first evidence implicating SIRT6 activity in controlling nuclear acetyl-CoA pools. The magnitude of acetyl-CoA changes that we observe in SIRT6 deficient cancer cells is comparable to those previously shown to have functional effects on histone acetylation and on cellular differentiation phenotypes. Thus, it will be interesting to determine whether SIRT6 also regulates ACLY and histone acetylation in other physiologic contexts, for example, in SIRT6-dependent aging-related or metabolic processes.

## Figures and Tables

**Figure 1 genes-12-01460-f001:**
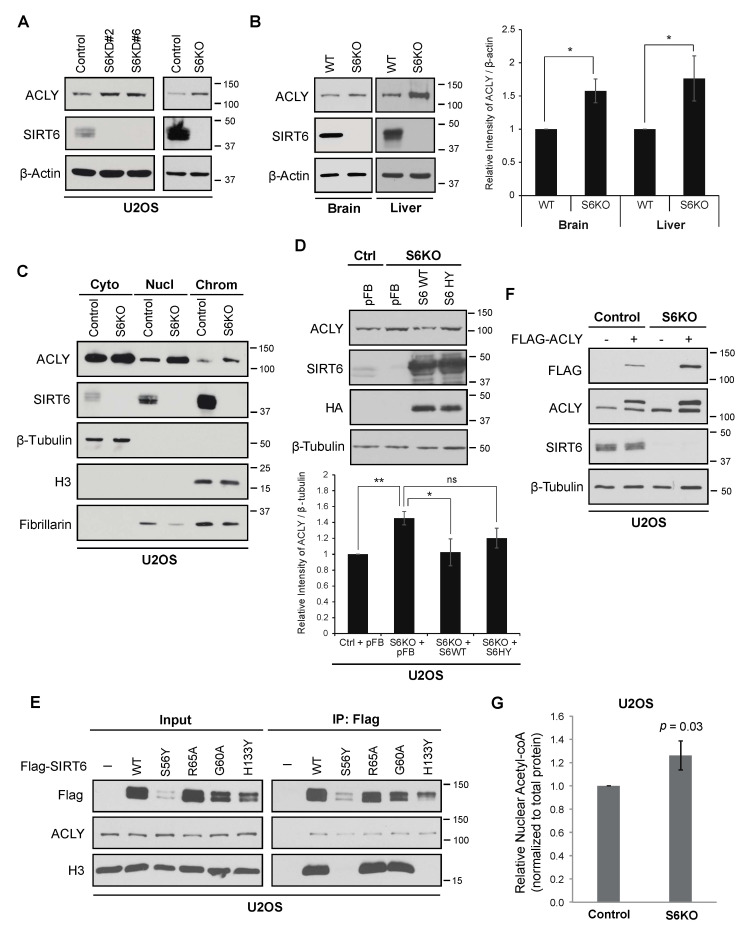
ACLY level increases with SIRT6 depletion. (**A**). Increased ACLY protein levels in SIRT6-deficient U2OS cells generated by knockdown (KD) or Crispr/Cas9-mediated knockout (KO). (**B**). Increased ACLY protein expression in SIRT6 KO mice tissues with quantification (Mean ± SEM of three experiments, * *p* < 0.05, one-tailed Student’s *t*-test). (**C**). Sub-cellular fractionation showing increased levels of nuclear and chromatin ACLY in SIRT6 KO cells compared to controls. β-tubulin, fibrillarin, and Histone H3 are shown as markers for the cytoplasmic, nuclear, and chromatin-enriched fractions. (**D**). Increased ACLY in SIRT6 KO cells reversed by overexpression of SIRT6 WT but not SIRT6 HY mutant with quantification (Mean ± SEM of three experiments, * *p* < 0.05; ** *p* < 0.01; ns: not significant, one-tailed Student’s *t*-test). pFB, empty vector control. (**E**). Co-IP of ACLY with Flag-tagged wild-type (WT) and catalytically inactive SIRT6 proteins overexpressed in U2OS cells. The S56Y and H133Y mutations abolish SIRT6 catalytic activity, whereas R65A and G60A have partially impaired deacetylase and ADP-ribosyl transferase activities. The Histone H3 levels show chromatin association of the SIRT6 proteins. (**F**). Increased abundance of Flag-ACLY and endogenous ACLY protein in SIRT6 KO cells. (**G**). Relative nuclear Acetyl-CoA levels in control and SIRT6 KO cells, normalized to total protein (Mean ± SEM of seven experiments, two-tailed Student’s *t*-test).

**Figure 2 genes-12-01460-f002:**
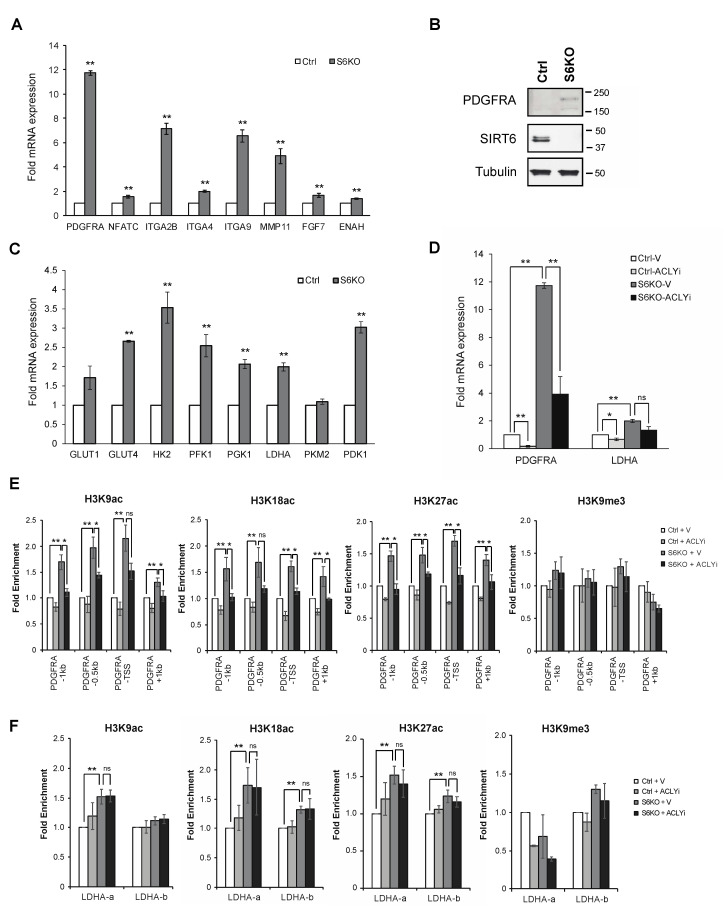
ACLY-dependent up-regulation of Acetyl-CoA responsive genes in SIRT6 deficient U2OS cells. (**A**). Expression of acetyl-CoA responsive genes in control and SIRT6 KO cells normalized to GAPDH (Mean ± SEM of four independent experiments, ** *p* < 0.01, two-tailed Student’s *t*-test). (**B**). Increased PDGFRA protein levels in SIRT6 deficient cells compared to control cells. (**C**). Expression of glycolytic genes in control and SIRT6 KO cells normalized to GAPDH (Mean ± SEM of three independent experiments, ** *p* < 0.01, two-tailed Student’s *t*-test). (**D**). Increased expression of PDGFRA but not LDHA in SIRT6 KO cells is attenuated by ACLY inhibitor (ACLYi, 50 uM; 24 h). Data are normalized to GAPDH and show Mean ± SEM of four independent experiments, * *p* < 0.05, ** *p* < 0.01, ns: not significant, two-tailed Student’s *t*-test. (**E**). Increased H3 acetylation marks on PDGFRA promoter regions by ChIP-qPCR, attenuated by ACLY inhibitor (Mean ± SEM of three independent experiments, * *p* < 0.05, ** *p* < 0.01, ns: not significant, two-tailed Student’s *t*-test). (**F**). Increased H3 acetylation marks on LDHA gene promoter regions by ChIP-qPCR, independent of ACLY inhibition (Mean ± SEM of three independent experiments, ** *p* < 0.01, ns: not significant, two-tailed Student’s *t*-test).

**Figure 3 genes-12-01460-f003:**
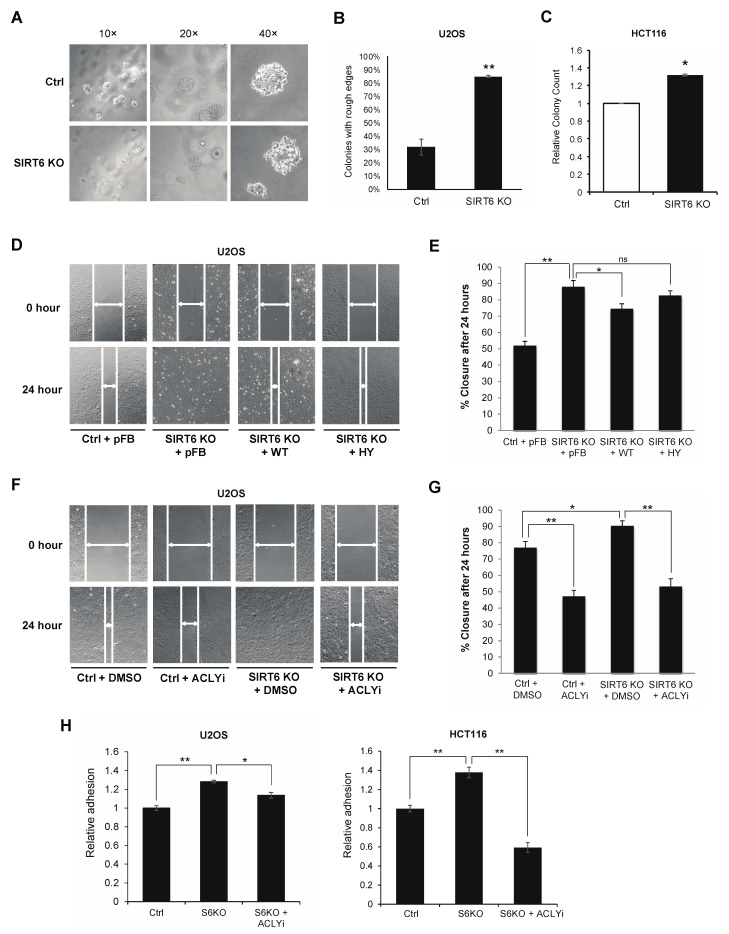
Adhesion and migration phenotypes of SIRT6-deficient cells and cells with ACLY inhibition. (**A**). Morphology of colonies formed by control or SIRT6 KO U2OS cells in soft agar colony formation assay. (**B**). Quantification of colonies with rough edge morphology. Colony counts were obtained from 10 random fields per well, four wells per cell type, *n* = 250, ** *p* < 0.01, two-tailed Student’s *t*-test. (**C**). Soft agar colony formation of control and SIRT6 KO HCT116 cells. Colony counts were obtained from three wells per cell type, * *p* < 0.05, two-tailed Student’s *t*-test. (**D**). Representative images of wound healing scratch assay in control and SIRT6 KO U2OS cells stably expressing empty vector (pFB), wild-type (WT), or H133Y mutant (HY) of SIRT6. Cells were grown to confluency then wounded, and wound closure was monitored every hour for 24 h. (**E**). Quantification of % wound closure after 24 h (Mean ± SEM of three independent experiments; * *p* < 0.05; ** *p* < 0.01; ns: not significant, two-tailed Student’s *t*-test). (**F**). Representative images of scratch assay in control and SIRT6 KO U2OS cells. Wound closure in the presence or absence of 50 µM of ACLY inhibitor (ACLYi) was followed over 24 h. (**G**). Quantification of % wound closure after 24 h (Mean ± SEM of three independent experiments, * *p* < 0.05, ** *p* < 0.01, two-tailed Student’s *t*-test). (**H**). Control or SIRT6 KO U2OS (left panel) and HCT116 (right panel) cell adhesion onto 1% fibronectin after 24 h treatment with vehicle or ACLY inhibitor (Mean ± SEM of three independent experiments, * *p* < 0.05, ** *p* < 0.01, two-tailed Student’s *t*-test).

**Figure 4 genes-12-01460-f004:**
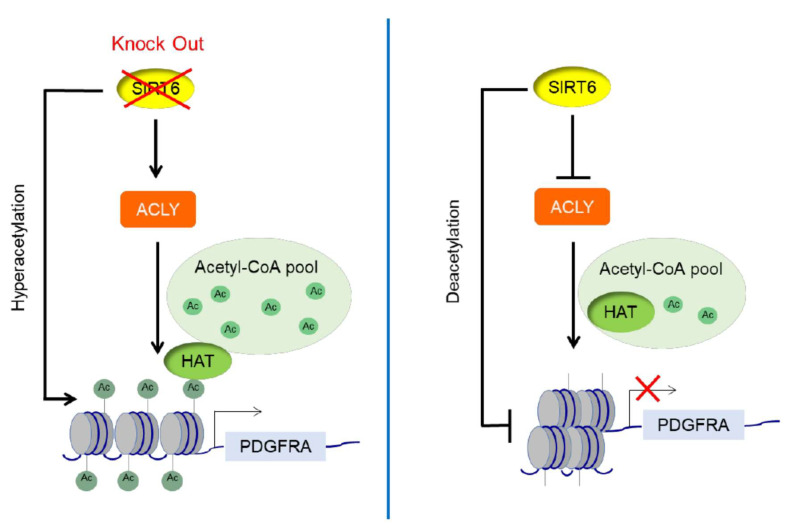
Model for the interplay between SIRT6 and ACLY, with a feed-forward loop to repress gene expression by coordinately regulating both histone deacetylation and availability of acetyl-CoA for histone acetylation.

## Data Availability

All data used in this paper are available in the article and Appendix A.

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
