# Peer review of "Mammalian SIRT6 Represses Invasive Cancer Cell Phenotypes through ATP Citrate Lyase (ACLY)-Dependent Histone Acetylation"

_genes, 2021, doi:10.3390/genes12091460_

Round 1
Reviewer 1 Report
In this work, Zheng and colleagues describe the functional interplay between ATP-citrate lyase (ACLY) and the histone deacetylase SIRT6, using human cancer cell lines. They show that SIRT6 ablation increases the total levels of nuclear ACLY and of the nuclear levels of the ACLY product Acetyl-CoA. As a consequence, SIRT6-KO cells show increased expression of several Acetyl-CoA-dependent genes associated to cell migration and ECM adhesion, and increased histone hyperacetylation at the promoter of one of them, PDGFRA. Finally, they show an increased invasive behavior of SIRT6-KO cells in vitro, that are reversed by overexpression of WT SIRT6 or with treatment with an ACLY inhibitor. The observations shown are clearly described, and add to the present literature on the mechanisms of SIRT6-mediated tumor suppression. However, there are a number of major remarks that need to be addressed:
General remarks:
- Indicate the exact cell type used for each panel. This is not clear form many panels in Figure 1, Supplementary Figure 1 or Figure 2, for example.
2.- When including Western blot experiments, several panels show a very dim difference between treatments/genotypes, but authors claim a clear difference in the main text that I do not see in the figures. Whenever differences are not dramatic, several replicates with precise quantification, should be presented and analyzed relative to internal controls (H3, actin tubulin or similar). I have included comments on some examples later on.
3.- Some panels (Figure 1D, Figure 2B, for example) are not presented following the order of the main text, and this makes the interpretation of the figures rather confusing. Please, present the data in the same order that in the text.
4.- Indicate the statistical methods used for each panel, both at the Figure legends and at Materials and methods section. Pay attention to using the appropriate statistical method, including corrections for multiple comparisons. This is particularly important for results with very dim differences, such as Figure 3E for example.
5.- Previous reports show that SIRT6 overexpression increased ACLY mRNA levels in a cell line different from the ones used here [53]. How are mRNA levels in SIRT6KO cells, compared with protein levels shown in Figure 1?
Panel-specific remarks
1.- Figure 1A: authors claim that SIRT6 downregulation induces increases in ACLY expression. However, in HCT116 cells, ACLY is not altered after SIRT6 elimination. Replicates of each condition should be included, and a quantification of all replicates normalized to internal control (b-actin) included.
2.- Something similar occurs in Figure 1B: the claimed increases in ACLY levels are not so evident in some cases (Pancreas or Brain), specially taking into account the b-actin control. Please, include at least 3 replicates (livers from 3 independent mice), and quantify the normalized band intensity to give a robust result.
3.- Figure 1E: authors claim that SIRT6-WT overexpression decreases total ACLY levels, but this is not apparent from the presented WB. Again, include 3 replicates per condition, a quantification and statistical analysis to have robust results.
4.- Supplementary Figure 1a: I do not see the difference in protein stability claimed by authors between control and S6KO. Again, several replicates should be treated and the average calculated and analyzed to claim this difference in protein stability.
5.- Figure 1D: I find the mild increase in PDGFRA in S6KO cells quite dim, not “dramatic” as indicated in the text. Again, please include replicates and quantify.
6.- There is no reference to Supplementary Figure 1B in the Results section. It is mentioned in Discussion, but the description and the point made with this panel are not clear to me. Also, this panel uses a different cell line from the ones used in the rest of the work. The information that ACLY-SIRT6 interaction is only apparent with overexpressed SIRT6, and not with endogenous SIRT6, using a different cell line than in Figure 1F and quite dirty bands (actually, there seems to be a dim band at SIRT6 height in the IP-ACLY lane) is very confusing. In my opinion, it should either be treated as a part of Results section, with all the appropriate explanations, or removed completely.
7.- Figure 2: gene expression is normalized to GAPDH as the sole housekeeping gene. However, this glycolysis gene can be strongly affected by metabolic stimuli, such as SIRT6-KO strategies. Please, include at least 2 other housekeeping genes less affected by metabolism than GAPDH, such as actin or b2-microglobulin.
8.- Figure 2B: the phrase in the main text: “We also compared the expression changes of the cell adhesion and migration genes to those of known SIRT6 target glycolytic genes in the same SIRT6-deficient cells (Figure 2B)” is confusing: in this panel we do not have cell adhesion and migration genes (shown in panel 2A), only glycolytic genes. Please clarify this sentence.
9.- Figure 2C: treatment with ACLYi significantly reduced the expression of LDHA in WT cells (Figure 2C), and changed significantly the expression of many SIRT6 target genes (Supplementary Figure 2), although authors claim that this treatment did not have any effect on these genes. Please modify the text and discuss these remarkable effects of ACLYi on SIRT6 target genes.
Also, please quantify the extent of Acetyl-CoA diminishment after ACLYi treatment, both in SIRT6-WT and in SIRT6-KO conditions, as shown in Figure 1G for SIRT6-KO cells.
10.- Figure 3A-C. What is the difference between the experiments shown in F3A-B and F3C? What is the relative colony count for U2OS, and the % of rough ended colonies in HT116 cells? It is strange that, for the same type of assay, different outcomes are shown for different cell types. Also, authors indicate that the soft agar colony formation was anchorage-independent, but in soft agar, colony formation depends on the agar anchorage. Please clarify.
11.- Figure 3D and 3F: it is confusing that upper panels (at 0 hours) show different treatments than lower panels (at 24 hours after scratching). Also, not all conditions quantified in Figures 3E and 3G are shown in these pictures: SIRT6KO+SITY6WT or SIRT6KO+SITY6HY at 24 hours are not shown, for example. Please, provide representative images at t0 and t24 for all conditions shown in quantifications to the right.
Also, what does “Luc” in the X axis of Figure 3E mean?
12.- Figure 3H: there are two different SIRT6KO clones shown for HCT116 cells, but only one for U2OS. Also, this is the first time that two clones are shown for HCT116 cells. Please, clarify why including these clones now, and not in previous panels.
As a resume, I consider the work presented here as potentially interesting to the specialist in the field, but a general revision of the manuscript should be performed to give more robustness to the presented results.
Reviewer 2 Report
Summary/main points: In this report authors shown that SIRT6 loss leads to increased levels of nuclear and chromatin-associated ACLY and nuclear acetyl-CoA level. This report stablishes a mechanistic link between SIRT6 and ACLY, to maintain nuclear acetyl-CoA pools in a coordinated manner and importantly, the expression of genes that contribute to the cell migration and adhesion phenotypes. This report reinforces the role of SIRT6 as a tumor suppressor gene not just as a histone deacetylase but also as suppressor of acetyl-CoA, precursor for histone acetylation. Both activities contribute to maintain repression of cancer cell gene expression programs. In addition, the authors provide the first evidence implicating SIRT6 activity in controlling nuclear acetyl-CoA pools, which in agreement with the authors, might have an impact in other physiologic contexts in which SIRT6 has been involved such as aging-related or metabolic processes. Both, rational and experimental design are very solid.
Minor concerns:
- In lines 378-9, authors state that SIRT6 deficiency increases ACLY protein levels post-transcriptionally. Loss of SIRT6 increases both cytoplasmic and nuclear ACLY protein levels (Figure 1). As the authors acknowledged, previous reports indicated that ACLY mRNA expression decreases upon miR122 cell treatment, which is an antagonist of SIRT6 [REF # 53]. Did the authors checked ACLY mRNA levels in the same cells and conditions used in Figure 1? These data should be included in supplementary information in order to maintain their claim.
- Quantification of Western-Blots shown in Figure 1 are missing. This is most important for Figure 1E, where the differences between SIRT6 KO and control conditions are mild. Please include at least Figure 1E quantification.
- In Figure 1F, authors performed a Co-IP of ACLY with Flag-tagged wild-type and catalytically inactive SIRT6 proteins. Even in the absence of quantification, seems apparent that SIRT6 interacts more with the S56Y mutant this is a very interesting finding and should be mentioned in the discussion section
- In line 252-2, authors state that ACLY inhibitor substantially reduced the upregulated expression of PDGFRA but had no effect on expression of the SIRT6-targeted glycolytic genes, such as LDHA (Figure 2C & S2). However, in Figure S2 some of the glycolytic genes tested seems to be affected by ACLY inhibition in SIRT6 KO cells with an apparently statistically significant increase e.g. GLUT4, HK2 & PDK1 (please provide all stats between treatments if not present). These observations deserve to be discussed in the text. Additionally, this brings to the next question, why did the authors choose LDLH as the example of a SIRT6 targeted gene? There are other genes that seem to be more up-regulated in SIRT6 KO cells such as HK2 & Please modify the text according to your answers.
- Finally, the manuscript would benefit from a model on the proposed mechanism of SIRT6-ACLY interplay in the repression of cancer cell gene expression programs. This would help to deliver the main message of the report and highlight the main claims of the work.
Other minor concerns:
- Figure S2 it is difficult to visualize. Could you change the pattern of the bar columns and make the bar legend bigger?
- Figures 4D and 4F are also difficult to visualize. Please be sure that the final figures have appropriate contrast.
